# Developmental constraints mediate the reversal of temperature effects on the autumn phenology of European beech after the summer solstice

Dominic Rebindaine[1,2]*, Thomas W Crowther[3,4], Susanne S Renner[5], Zhaofei Wu[2,6], Yibiao Zou[1,2], Lidong Mo[7], Haozhi Ma[2], Raymo Bucher[1], Constantin M Zohner[3,8]

[1]Department of Environmental Systems Science, ETH Zurich (Swiss Federal Institute of Technology), Zurich, Switzerland; [2]WSL Swiss Federal Institute for Forest, Snow and Landscape Research, Birmensdorf, Switzerland; [3]BRANCH Institute, Zug, Switzerland; [4]King Abdullah University of Science and Technology (KAUST), Thuwal, Saudi Arabia; [5]Department of Biology, Washington University, Saint Louis, United States; [6]College of Water Sciences, Beijing Normal University, Beijing, China; [7]Department of Plant Biology and Ecology, College of Life Science, Nankai University, Tianjin, China; [8]Institute for Future Initiatives, The University of Tokyo, Tokyo, Japan

*For correspondence:
dominic.rebindaine@usys.ethz.ch

## eLife Assessment

This article presents **valuable** findings on how the timing of cooling affects autumn bud set in European beech saplings. The study leverages extensive experimental data and provides an interesting conceptual framework for the various ways in which warming can affect bud set timing. The statistical analysis is very well considered, while indicating some factors that may temper the authors' claims. The factorial experiments offer **solid** support.

**Abstract** Accurate projections of temperate tree growing seasons under climate change require representing developmental constraints that determine tree resource allocation. A phenological 'switch point' after the summer solstice (21 June) has been proposed, with pre-solstice warming advancing autumn phenology and post-solstice warming delaying it. We propose that this switch is flexible and occurs at the compensatory point between early-season development and late-season temperature effects. We performed trans-solstice climate manipulation experiments on potted European beech (*Fagus sylvatica*) saplings to test (i) how spring leaf-out timing and June-August temperatures influence end-of-season timing (bud set and leaf senescence) and (ii) whether daytime and nighttime temperatures before and after the solstice have distinct effects. Bud set and senescence were tightly coupled ($R^2$=0.49), with stronger bud responses. Each day of delayed leaf-out postponed bud set by 0.24±0.06 days and senescence by 0.22±0.08 days. July full-day cooling delayed autumn phenology in late-leafing individuals (bud set +4.9±2.6 days; senescence +3.1± 2.8 days) but had a negligible impact on early-leafing trees. August full-day cooling advanced both stages. Pre-solstice daytime cooling had no effect, while post-soltice daytime cooling advanced autumn phenology. Nighttime cooling consistently delayed bud set. These findings support the Solstice-as-Phenology-Switch model and highlight the central role of developmental progression in constraining growing seasons. Faster early-season development – especially under nighttime warming – moves trees past the switch earlier, increasing sensitivity to late-season cooling and thereby triggering earlier autumn phenology.

Phenology models should incorporate these developmentally-mediated and diel-specific temperature responses.

## Introduction

Climate shifts are leading to rapid, species-specific changes in phenology and ecosystem productivity (*Boisvenue and Running, 2006*; *Menzel et al., 2006*; *Thackeray et al., 2016*; *Yang and Rudolf, 2010*). In temperate forests, changes in the timing of spring leaf-out, autumn leaf-senescence, and bud set are modifying water, energy, and carbon cycles (*Peñuelas et al., 2009*; *Richardson et al., 2013*), with extended growing seasons increasing net ecosystem carbon uptake by up to 9.8 gC m$^{-2}$ day$^{-1}$ (*Keenan et al., 2014*). Therefore, understanding the interaction between climate change and temperate forest phenology is pivotal to improving forecasts of community dynamics and carbon sequestration.

The past few decades have seen delays in the onset of temperate autumn phenology, but these changes are much smaller in magnitude compared to the advances in spring leaf-out observed during the same period (*Gill et al., 2015*; *Piao et al., 2019*). This is unexpected, given that experiments have demonstrated a high sensitivity of leaf senescence to autumn warming, with phenological responses even surpassing the temperature sensitivity (days per °C) of spring leaf-out (*Fu et al., 2018*). One possible explanation for this discrepancy is that other factors may counterbalance the effects of autumn warming, with some studies finding that earlier leaf-out leads to earlier leaf senescence (*Fu et al., 2014*; *Keenan and Richardson, 2015*; *Zani et al., 2020*). This connection between spring and autumn phenophases could be due to developmental and nutrient constraints that affect carbon source-sink dynamics in temperate trees (*Paul and Foyer, 2001*; *Zani et al., 2020*; *Zohner et al., 2023*; *Gessler and Zweifel, 2024*), imposing limits on the phenological growing season.

The Solstice-as-Phenology-Switch hypothesis, supported by experiments and observational data, posits that autumn phenology is driven by two counteracting temperature effects (*Zohner et al., 2023*). From the start of the growing season, warmer air temperatures drive faster development and growth activity, such as tissue hardening, meristematic activation, cell division and maturation, allowing trees to fulfil their developmental requirements more quickly and initiate leaf senescence earlier (*Körner, 2021*; *Körner et al., 2023*; *Tumajer et al., 2021*). The majority of activity occurs well before the end of the phenological growing season (*Cruz-García et al., 2019*; *Etzold et al., 2022*; *Körner et al., 2023*), so this effect is expected to act primarily during the early-season (henceforth referred to as the early-season developmental [ESD] effect). Conversely, the rate at which senescence progresses is predominantly mediated by temperature (*Estiarte and Peñuelas, 2015*; *Fu et al., 2018*; *Zohner et al., 2023*), with cooling triggering senescence progression and thus, warmer temperatures slowing senescence and delaying the end of the growing season (henceforth referred to as the late-season temperature [LST] effect). Year-round warming will, therefore, exert non-linear effects, with early-season cooling delaying autumn phenology (ESD effect), while late-season cooling advances it (LST effect). This seasonal reversal in temperature responses provides a mechanistic explanation for why shifts in autumn phenology are typically smaller and less consistent in direction than those observed for spring leaf-out (*Piao et al., 2019*). The fact that this reversal occurs after the solstice points to the role of photoperiod in regulating plant physiology (*Bauerle et al., 2012*; *Petterle et al., 2013*; *Singh et al., 2017*). However, the realised timing of this reversal is context-specific and appears to have advanced in recent decades (*Zohner et al., 2023*).

To explain this flexibility, we propose that the solstice acts as an environmental switch, with declining daylength providing a consistent and biologically meaningful cue that initiates the LST effect, while the ESD effect can persist beyond the solstice depending on an individual's developmental state in a given year. Under this framework, the reversal in temperature responses occurs at a compensatory point where the advancing ESD effect is balanced by the delaying LST effect (*Figure 1*). This point should, therefore, not be fixed to a calendar date but instead vary with developmental progression each year.

For temperate trees, developmental outcomes are the production of viable seeds, cessation of primary and secondary growth, and maturation of perennating tissues, including leaf and flower buds (i.e. completion of bud set) before the onset of frost (*Cooke et al., 2012*; *Rohde and Bhalerao, 2007*; *Tanino et al., 2010*). Because environmental conditions constrain the rate and duration of

**eLife digest** Many plants adjust their routines with the seasons. In regions with warm summers and cold winters, trees must balance growing for as long as possible while avoiding frost damage. Deciduous trees – those that shed their leaves each year – prepare for winter by stopping growth, strengthening protective tissues and eventually dropping their leaves.

The timing of these changes is called autumn phenology, which refers to plant processes that occur at the end of the growing season. Two key stages mark this process: bud set, when future leaf buds stop growing, and leaf senescence, when leaves break down nutrients, change color, and are eventually shed.

To time these changes correctly, trees rely on a combination of external and internal signals. External cues include temperature and day length, while internal cues reflect the tree's developmental state, such as leaf age or the number of life-cycle events already completed during the year.

Recent work has shown that the summer solstice on 21 June acts as a phenological 'switch point', with pre-solstice warming advancing autumn phenology and post-solstice warming delaying it.

To understand why autumn phenology in temperate trees responds differently to temperature changes before and after the summer solstice, Rebindaine et al. tested how trees respond to cooling at different times during the season.

They studied young European beech trees, an important forest species in Europe. In spring, they cooled half of the trees while keeping the others under normal conditions, creating groups that developed at different rates. They then exposed both groups to cooler temperatures in July and August. Cooling in July delayed autumn timing only in the slower-developing trees, showing that a tree's developmental stage strongly influences its response. In contrast, cooling in August caused all trees to prepare for autumn earlier, regardless of developmental differences, indicating that late-summer temperature cues override internal variation.

Improving our understanding of how temperate deciduous trees control their phenology enables scientists to better predict how trees interpret environmental signals and how forests will respond to climate change. Incorporating this knowledge into ecological models will provide more accurate forecasts of growing-season length and carbon uptake. This, in turn, could help guide decisions about which tree species to plant and how forests should be managed sustainably. Further studies across additional species will be needed to determine how broadly these mechanisms apply.

development, the timing of this compensatory point should vary among years and individuals. Under more advanced early-season development, this point is expected to be reached shortly after the solstice, whereas under slower development, it occurs later (*Figure 1*). However, such flexibility has yet to be demonstrated for processes directly linked to growth cessation, such as autumn bud set.

In this study, we investigate how air temperature changes around the summer solstice affect end-of-season timing in European beech (*Fagus sylvatica* L.), using bud set and leaf senescence as key physiological markers of autumn phenology (*Mariën et al., 2021*; *Singh et al., 2017*; *Zohner and Renner, 2019*). To elucidate the underlying mechanisms, we employ strong physiological forcing experiments specifically designed to maximise signal-to-noise ratios rather than representing contemporary or future climates. We ask three main questions: (i) How does cooling before vs. after the solstice affect end-of-season timing? (ii) How do diel (daytime vs. nighttime) temperature changes differ in their effects? (iii) Does the timing of leaf-out modulate the reversal of temperature effects on autumn phenology? Diel temperature variations are relevant because trees typically grow more at night when temperatures and water deficit are lower (*Mencuccini et al., 2017*; *Steppe et al., 2015*; *Zweifel et al., 2021*). Thus, developmental effects mediated by temperature may be especially pronounced during nights. Moreover, long-term trends in climate show asymmetries between daytime and nighttime warming (*Vose et al., 2005*; *Zhong et al., 2023*). To address the third question, we experimentally manipulated spring conditions to create early-leafing and late-leafing individuals within the same growing season, allowing us to assess whether slowed early-season development postpones the point at which late-season cooling advances autumn phenology (*Figure 1*). As trees continuously adapt their physiological responses to environmental conditions over time, we expected their temperature responses to differ both across months (June-August) and between day and night.

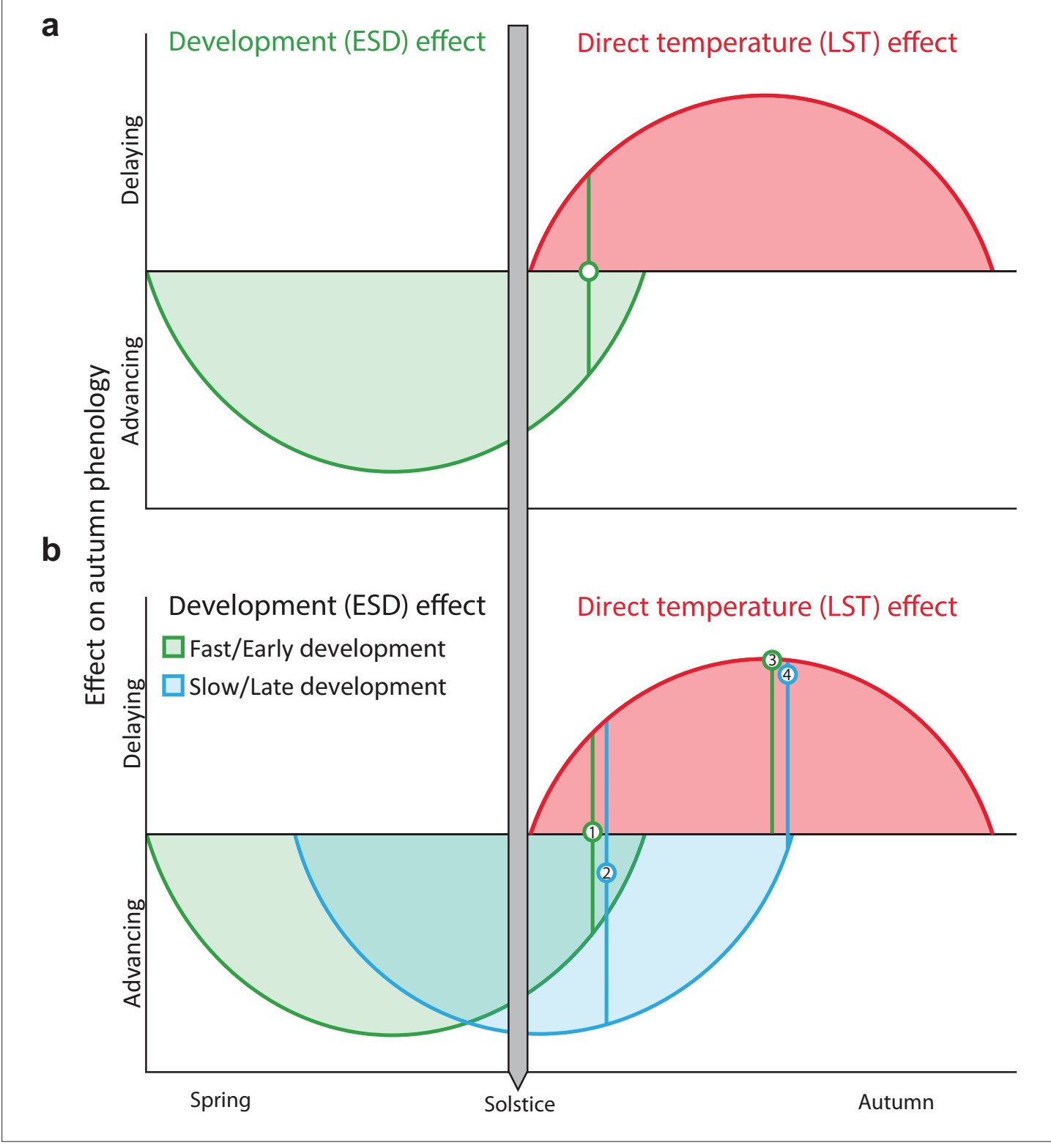

**Figure 1.** Conceptual model of autumn phenological responses of temperate trees to early-season development (ESD) and late-season temperature (LST) effects. Autumn phenology, represented in this study by the timing of primary growth cessation (bud set) and leaf senescence (50% loss of leaf chlorophyll content), is influenced by two opposing factors: ESD and LST. (**a**) In our model, ESD, which is driven by temperature, has an advancing effect on autumn phenology that lasts until shortly after the summer solstice (green curve). Higher temperatures cause trees to complete their annual life cycles faster, allowing them to set buds and senesce leaves. After the summer solstice, as days shorten, trees become increasingly sensitive to

*Figure 1 continued on next page*

*Figure 1 continued*

cooling conditions, so late-season warming slows the progression of bud set and senescence, delaying autumn phenology (LST effect, red curve). The compensatory point, where the advancing ESD effect is balanced by the delaying LST effect, is represented by the green circle. (**b**) According to the model, the position of this compensatory point is flexible and varies between years based on the speed of development. When development is slow or starts late (blue curve), the compensatory point is reached later than under fast or early development (green curve). Therefore, shortly after the solstice, the effect of temperature on autumn phenology differs between fast/early and slow/late developing individuals. For example, point 1 (green circle) shows no net temperature effect on phenology in fast/early trees. By contrast, point 2 (blue circle) shows that in slow/late developing trees, warmer temperatures shortly after the solstice (in July) still advance autumn phenology. However, as the growing season progresses and days shorten, trees become more responsive to cooling regardless of prior developmental speed (LST effect strengthens). Additionally, a weakening in the ESD effect is expected as individuals approach completion of their developmental requirements. By August, both fast/early and slow/late trees should, therefore, exhibit similar phenological responses, with warming consistently delaying autumn phenology (points 3 and 4).

## Materials and methods
### Selected species

*Fagus sylvatica* L. has a large temperate European distribution, is highly economically and ecologically important, and is likely threatened by climate change (*Gessler et al., 2006*). Therefore, improving our understanding of how *F. sylvatica*'s growing season is modulated is vital to the management of European forests. Moreover, *F. sylvatica* has frequently been used as a key species to study temperate tree phenology (*Dittmar and Elling, 2006*; *Fu et al., 2018*; *Mariën et al., 2025*; *Švik et al., 2025*; *Vitasse et al., 2011*; *Zani et al., 2020*; *Zohner et al., 2023*; *Zohner and Renner, 2019*).

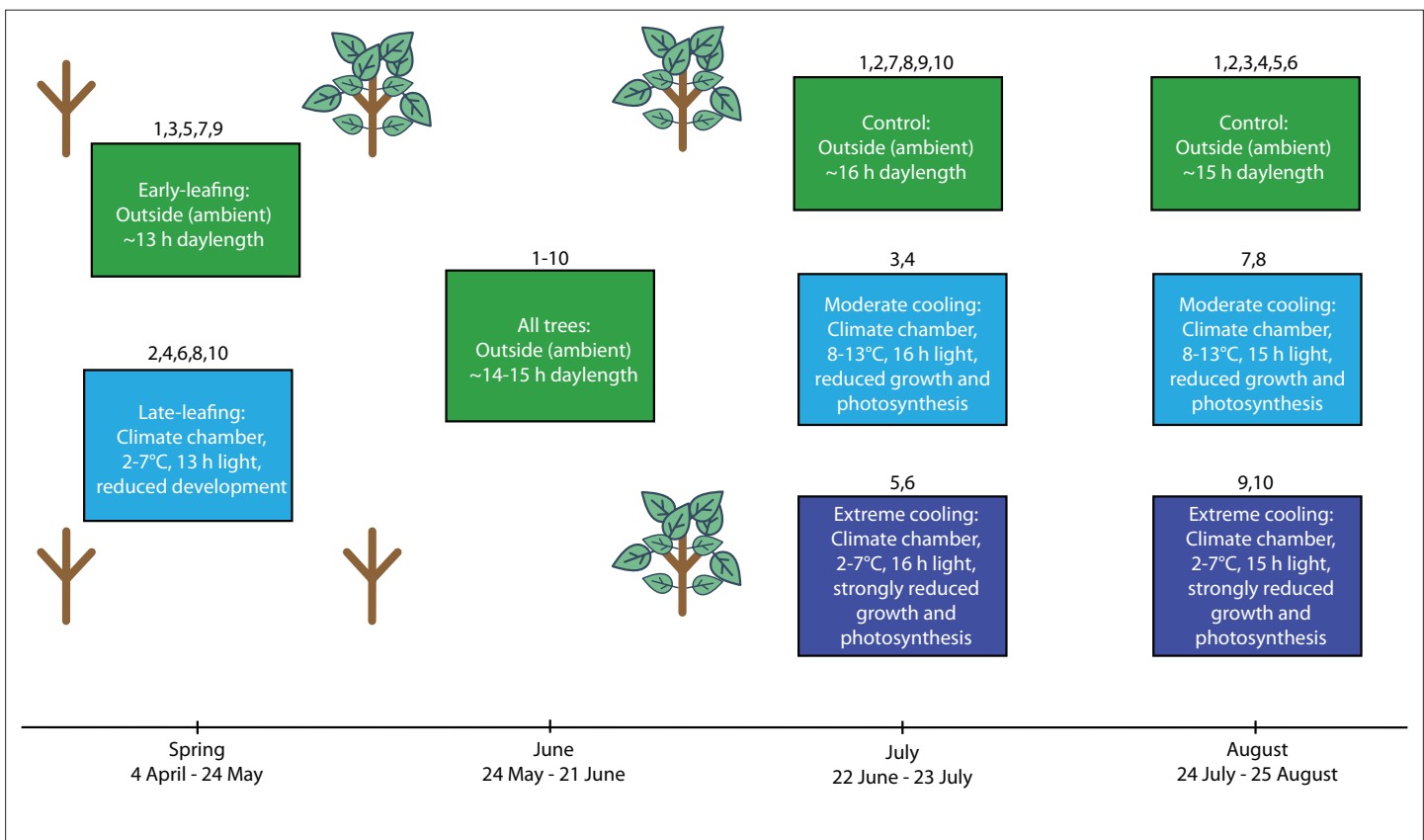

**Figure 2.** Depiction of the experimental timeline and settings for experiment 1. Each box corresponds to a specific treatment block at that point in time. The numbers inserted above the boxes refer to the treatment group (see *Table 1* for details). Each box contains information on the location of the trees, the specific conditions they were under and the intended physiological effects of those conditions. The sapling graphics highlight differences in early-season developmental progression for the early-leafing and late-leafing groups. Following the August treatments, all trees were placed outside under ambient conditions in a randomised block design.

## Experiment 1

To test the antagonistic influences of the ESDE and the LSTE on autumn phenology, we set up an experimental population (n=267) of 40–60 cm tall European beech (*F. sylvatica*) trees in Zurich, Switzerland in 2023. The trees were sourced from a local nursery, and each tree was placed individually in a 20 L plastic pot containing a 1:1:1 sand/peat/organic soil mixture with a Nitrogen (N) concentration of ~65 g m$^{-3}$, a Phosphate ($P_2O_5$) concentration of ~140 g m$^{-3}$, and a Potassium ($K_2O$) concentration of ~400 g m$^{-3}$. Individual trees were assigned randomly to one of 10 treatment groups (26≤n ≤ 27). Treatment groups were cooled at different times of the year, using different cooling levels (*Figure 2*, *Table 1*). To arrest spring development and thereby generate early-leafing and late-leafing individuals, half of the experimental population was placed outside under ambient conditions, while the other half was cooled in climate chambers from 4 April to 24 May. The chambers were set to a low of 2 °C at night and a high of 7 °C during the day, following a simulated day-night cycle of temperature and light availability (13 hr photoperiod at ~4300 lux). These cold conditions were not intended to mimic natural European conditions but to strongly slow development without causing damage. Temperatures below 10 °C substantially limit development and growth by slowing cell division and mitotic activity, with rates approaching zero as temperatures near 0 °C (*Körner, 2021*; *Tumajer et al., 2021*). From 24 May to 21 June, all trees were kept outside under ambient conditions in a randomised block design. All trees were monitored to observe their individual leaf-out dates, which was defined as the date when >50% of their leaves had unfolded, corresponding to BBCH15 (*Capdevielle-Vargas et al., 2015*).

During the summer treatments, control trees were placed outside under natural ambient conditions. We did not use climate chambers for the control groups because warm conditions in these chambers can introduce unique physiological stressors—such as aphid infestations—not present in cold chambers (*Bezemer et al., 1998*). In experiment 2, however, we included an additional chamber control to test for potential chamber effects. Although we observed higher aphid abundances in these warm control chambers, chamber exposure itself had no detectable effect on autumn phenology. This strengthens the inference that the phenological shifts observed in our treatments reflect the intended temperature manipulations. Nevertheless, for experiment 1, where chamber controls were not included, we cannot entirely rule out the possibility that chamber conditions exerted some unintended influence on treated trees compared with ambient controls.

The July treatments took place directly after the summer solstice from 22 June to 23 July. Treated trees were placed in climate chambers set to a low of 2 °C at night and a high of 7 °C during the day or a low of 8 °C at night and a high of 13 °C during the day depending on their cooling level (extreme and moderate, respectively). Trees in the chambers experienced a photoperiod of 16 hr at ~7300 lux. The extreme cooling was designed to severely impair cell division, expansion, and maturation as described above. Additionally, under this temperature regime, photosynthesis should be reduced by >40% (*Körner, 2006*). The moderate cooling regime, which was still much colder than the ambient conditions, should have also impaired growth and development as well as reduced photosynthesis by at least 30% (*Körner, 2006*; *Körner, 2021*; *Lenz et al., 2014*). These settings aimed to generate large temperature differences between treatments and maximise our ability to detect the mechanism underlying the solstice switch, rather than to represent naturally occurring conditions. The August treatments took place from 24 July to 25 August under the same conditions as the July treatments except with a 15 hr photoperiod. For the remainder of the experiment, all trees were kept outside in a randomised block design. Throughout the experiment, all trees were watered frequently to ensure constant water supply (*Figure 4—figure supplement 1*).

To observe the effects of our treatments on the development of overwintering buds, we monitored bud growth to derive bud set dates, our marker for the cessation of primary aboveground development and the beginning of autumn phenology for all trees. On each tree, the terminal bud of the primary shoot and the terminal bud on a random lateral stem were selected and tagged for measurement. Each selected bud was measured to 0.01 mm precision using a digital calliper (CD-P8"M, Mitutoyo Corp, Japan). We measured all buds on a regular basis from 4 July to 2 November. Bud set was defined as the date when each bud reached 90% of its own maximum length, which is considered to be an indicative stage of aboveground primary growth cessation (*Signarbieux et al., 2017*; *Zohner and Renner, 2019*). Bud lengths were linearly interpolated between measurement dates to derive the date at which they reached the 90% threshold.

**Table 1.** Description of the temperature treatments applied in experiment 1.
July treatments were from 22 June to 23 July. August treatments were from 24 July to 25 August. Ambient means outside under natural conditions. Temperature ranges indicate the daily minimum and maximum temperatures experienced by trees inside climate chambers. The early-leafing and late-leafing trees were experimentally generated by placing potted trees in climate chambers from 4 April to 24 May and cooling them to 2 °C at night and 7 °C during the day to arrest spring development.

| Treatment | 1 | 2 | 3 | 4 | 5 | 6 | 7 | 8 | 9 | 10 |
|---|---|---|---|---|---|---|---|---|---|---|
| Leaf-out timing | Early | Late | Early | Late | Early | Late | Early | Late | Early | Late |
| Treatment timing | – | – | July | July | July | July | August | August | August | August |
| Cooling level | Control (ambient) | Control (ambient) | Moderate (8–13°C) | Moderate (8–13°C) | Extreme (2–7°C) | Extreme (2–7°C) | Moderate (8–13°C) | Moderate (8–13°C) | Extreme (2–7°C) | Extreme (2–7°C) |

To obtain a more holistic perspective of autumn phenology, we also derived leaf senescence dates from leaf spectral index measurements taken using a SPAD-502 Plus (Soil Plant Analysis Development, Minolta Camera Co., Ltd, Tokyo, Japan). We measured nine leaves per individual (three each from the top, middle, and bottom of the crown) on a monthly basis in summer, every other week in September, and on a weekly basis from October to mid-December. We removed measurements taken on 7 December from the analysis as they were unreasonably high, likely due to false readings caused by extensive leaf browning. SPAD readings were then converted to total chlorophyll content (*Chl* in µg/g fresh weight) using an empirically established equation for *Fagus sylvatica* leaves (*Percival et al., 2008*):

$$Chl = -0.0029 \times SPAD^2 + 1.175 \times SPAD + 3.8506.$$

The chlorophyll content between two consecutive measurement dates was estimated using linear interpolation. For each treatment and measurement date, we removed individual chlorophyll measurements that were more than 1.5x the interquartile range below the lower quartile or above the upper quartile. As an additional cleaning step, we completely removed any trees that had more than one data point removed in the last step. Finally, we calculated individual leaf senescence dates as the day-of-year when chlorophyll content last fell below 50% of the observed peak chlorophyll content.

## Experiment 2

To observe the effects of pre- and post-solstice daytime and nighttime temperature on autumn phenology, we set up an experimental population (n=180) of four-year-old *Fagus sylvatica* trees in Zurich, Switzerland in 2022. The trees were sourced from a local nursery and each tree was placed

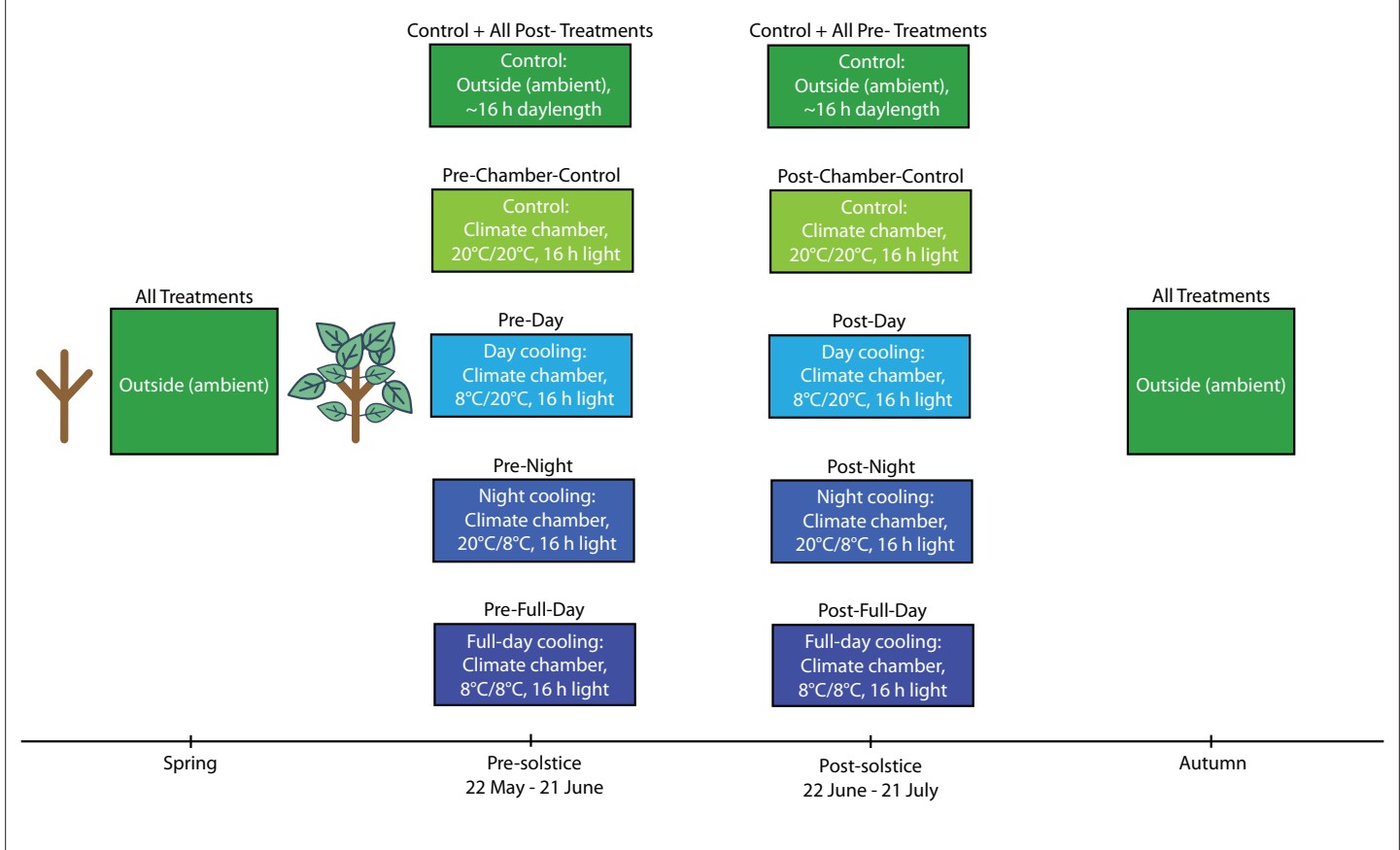

**Figure 3.** Depiction of the experimental timeline and settings for experiment 2. Each box corresponds to a specific treatment block at that point in time. The text inserted above the boxes refers to the treatment group (see *Table 2* for details). Each box contains information on the location of the trees and the specific conditions they were under. Temperature regimes are given in the format day/night. The sapling graphics indicate that all trees had equal opportunities for early-season development.

**Table 2.** Description of the temperature treatments applied in experiment 2.

June treatments were from 22 May to 21 June. July treatments were from 22 June to 21 July. Ambient means outside under natural conditions. The remaining temperature regimes are in the format day/night and refer to the temperatures applied to trees in climate chambers.

| Treatment | Control | Pre-chamber control | Pre-day | Pre-night | Pre-full-day | Post-chamber control | Post-day | Post-night | Post-full-day |
|---|---|---|---|---|---|---|---|---|---|
| Cooling time of year | - | June | June | June | June | July | July | July | July |
| Cooling time of day | - | - | Day | Night | Full-day | - | Day | Night | Full-day |
| Temperature regime | Ambient | 20/20 °C | 8/20 °C | 20/8 °C | 8/8 °C | 20/20 °C | 8/20 °C | 20/8 °C | 8/8 °C |

individually in a 20 L plastic pot containing a 1:1:1 sand/peat/organic soil mixture with a Nitrogen (N) concentration of ~65 g m$^{-3}$, a Phosphate ($P_2O_5$) concentration of ~140 g m$^{-3}$, and a Potassium ($K_2O$) concentration of ~400 g m$^{-3}$.

The ambient control treatment consisted of 36 trees exposed to natural ambient conditions. The remaining eight treatments were each applied to 18 trees and included cooling in climate chambers with simulated ambient day length (16 hr) and light intensity (~6900 lux). The pre-solstice cooling treatments were applied between 22 May and 21 June. The post-solstice cooling treatments were applied between 22 June and 21 July. The pre- and post-solstice treatments included four levels each: Chamber control, where the trees were continuously subjected to 20 °C; Day cooling, where trees were subjected to 8 °C in the daytime and 20 °C at night; Night cooling, where trees were subjected to 20 °C in the day and 8 °C at night; Full-day cooling, where trees were continuously subjected to 8 °C (*Figure 3*, *Table 2*). Due to warm temperatures, chamber control trees were subject to an increase in aphid abundance; however, this did not alter their bud set timing (see data analyses). After treatment, all trees were placed in a randomised block design outside under ambient conditions. Soil moisture content was regulated by frequent watering. We measured all buds on a weekly basis from 25 August to 3 November following the same methodology as in experiment 1. We also measured leaf-level $CO_2$ assimilation rates during the pre-solstice treatment window (see *Zohner et al., 2023* for methodology), and derived leaf senescence dates following experiment 1.

## Data analyses

For experiment 2, we performed one-way ANOVAs that showed no significant differences between the end-of-season dates of the ambient, pre-chamber, and post-chamber control groups (for bud set: $F_{2,136} = 0.346$, p=0.708, see *Figure 5—figure supplement 1*; for leaf senescence: $F_{2,63} = 0.931$, p=0.399). Therefore, for the following analyses, we treated all three control treatments as one control treatment with 72 trees. For both experiments, we ran linear models using treatment and bud type (apical vs. lateral) as predictors and bud set day-of-year, absolute bud growth, or relative bud growth as the response variable. When modelling leaf senescence day-of-year, we used treatment as the sole predictor. For experiment 1, effect sizes were calculated in comparison to the corresponding control treatment, i.e., all early-leafing treatments were compared to the early-leafing control treatment and all late-leafing treatments were compared to the late-leafing control. Absolute and relative bud growth were calculated as:

$$Absolute = max\ length - min\ length$$

$$Relative = \left(1 - \frac{min\ length}{max\ length}\right) * 100$$

Where:

*min length* = the first measured bud length,

*max length* = the measured bud length when the bud first surpassed 90% of its final length.

To determine the sensitivity of autumn bud set timing to spring leaf-out timing, we ran a linear mixed-effects model using leaf-out day-of-year and bud type as the fixed predictor variables, summer temperature treatment as a random effect and bud set day-of-year as the response variable. We ran the same analysis for leaf senescence timing without the bud type effect.

To quantify the relative contributions of the ESDE and the LSTE to variation in bud set timing in experiment 1, we ran a variance partitioning analysis. We fit a linear model with leaf-out day-of-year, summer cooling treatment (combination of cooling timing and level, e.g. July_Moderate) and bud type as explanatory variables. Variance partitioning was conducted using the *varpart* function in the *vegan* R package (*Oksanen et al., 2025*), which decomposes explained variance into unique and shared components. Additionally, we calculated the intra-class correlation coefficient (ICC) to compare inter-individual (within treatment) vs between treatment variance (*Nakagawa et al., 2026*).

$$ICC = \frac{V_T}{V_T + V_W}$$

Where $V_T$ = between treatment variance and $V_W$ = within treatment variance.
All statistical analyses were performed in R version 4.5.1 (*R Development Core Team, 2025*).

## Results
### Experiment 1
### ESD effect
Linear modelling showed that across all treatments, late-leafing trees cooled during spring delayed their bud set by 4.6±1.3 days (mean ± 2 SE, *p*<0.01). Across all treatment pairings, the mean bud set date for the late-leafing group always occurred later than for the early-leafing group (*Figure 4—figure supplement 2*). A linear mixed effects model, including leaf-out day-of-year and bud-type as fixed effects and treatment as a random effect revealed that, on average, each day delay in spring leaf-out was associated with a delay of 0.24±0.06 days in bud set timing. Across all treatments, bud type had a small effect, lateral buds set on average 1.1±1.2 days earlier than apical buds (*p*=0.06). When modelling only the early-leafing groups, bud type had no discernible effect (0.04±1.64 days earlier for lateral buds, *p*=0.96). However, when modelling only the late-leafing groups, lateral buds set earlier than apical buds (2.17±1.64 days advancement, *p*=0.01).

### LST effect
The effect of July cooling differed strongly between the early- and late-leafing trees: Within the early-leafing group, moderate cooling in July led to a small, non-significant delay in bud set compared to the ambient July treatment (1.4±2.6 days, *p*=0.29). July cooling had a much greater impact on late-leafing trees, leading to a 4.8±2.6 day delay in bud set compared to the ambient July treatment in the late-leafing group (*p*<0.01).

In contrast to July cooling, the effects of August cooling did not differ between early- and late-leafing trees, advancing bud set in all treatment groups (*Figure 4b*). Cooling in August led to a 4.5±2.6 day (*p*<0.01) and a 4.4±2.6 day (*p*<0.01) advancement in bud set for early- and late-leafing trees, respectively. The extreme cooling treatments showed similar patterns, although slightly less clearly (*Figure 4—figure supplement 3*).

Variance partitioning revealed that the ESD effect (leaf-out day-of-year) explained 8.8% of the total variance in bud set timing, the LST effect (summer cooling treatment) explained 14.6%, and bud type explained 0.3%. Shared variance values were ≈ 0%, indicating no meaningful overlap. The intra-class correlation coefficient was 0.26, suggesting high levels of inter-individual (within treatment) variance. Effects of absolute and relative growth on bud set date can be found in *Figure 4—figure supplements 4 and 5*. Bud set timing had no effect on final bud length (*Figure 4—figure supplement 6*).

## Experiment 2
Linear modelling showed that pre-solstice full-day (day and night) cooling delayed autumn bud set by 4.1±3.6 days (mean ± 2 SE, *p*=0.02) (*Figure 5*). Pre-solstice nighttime cooling had a similar effect, delaying bud set by 4.2±3.6 days (*p*=0.02). By contrast, pre-solstice daytime cooling had no significant effect on bud set (0.1±3.7 days, *p*=0.94). Post-solstice full-day cooling advanced autumn bud set by 5.2±3.6 days (*p*<0.01). Similarly, post-solstice daytime cooling advanced bud set by 5.3±4.3 days

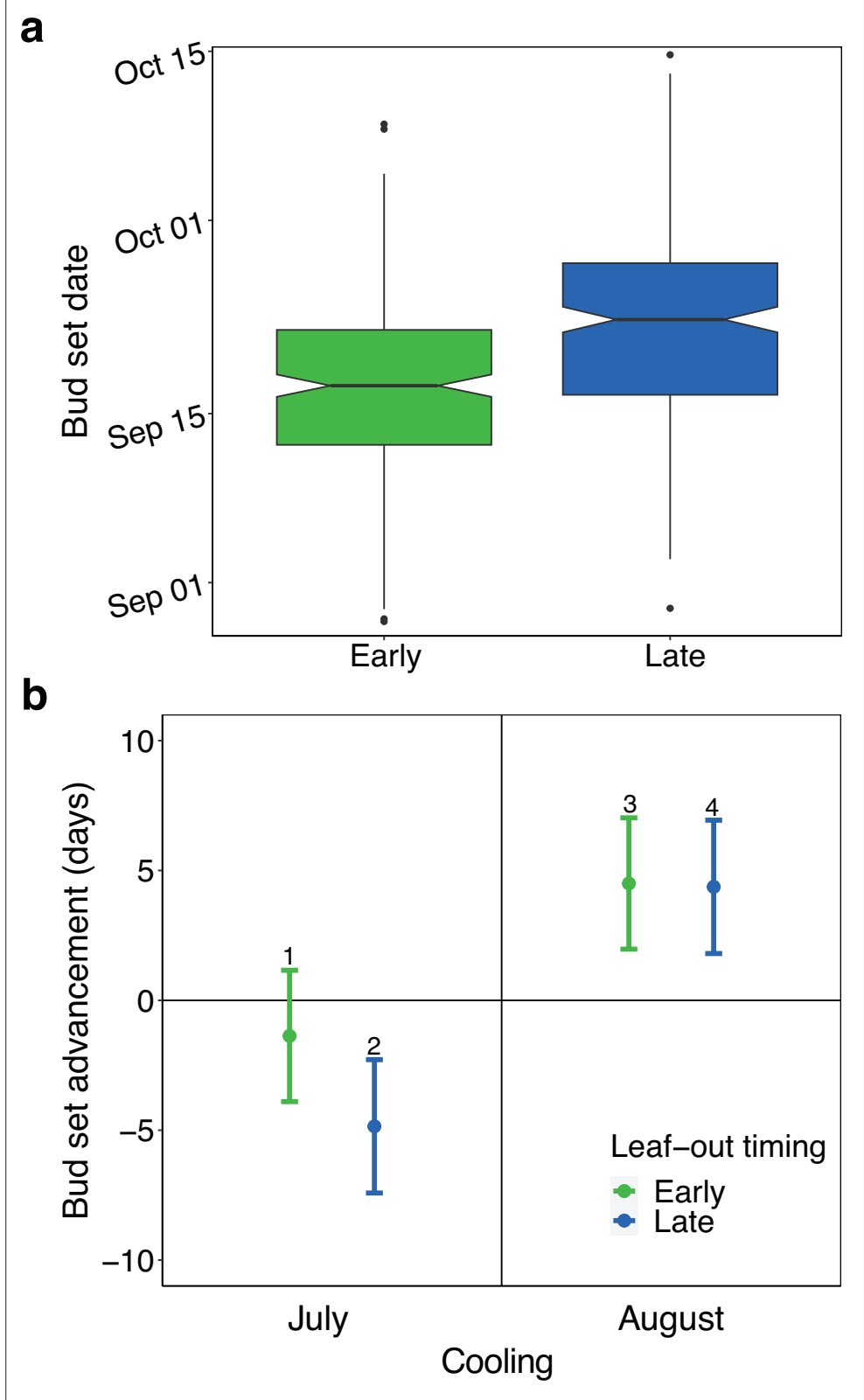

**Figure 4.** Effects of early-season development and late-season temperature on the timing of autumn bud set in *Fagus sylvatica* (experiment 1). (**a**) Bud set dates for early- (green) and late- (blue) leafing trees, including all treatments. Late-leafing trees were cooled (2–7°C) in climate chambers from 4 April to 24 May to arrest their development and delay leaf-out. (**b**) Effects of July (22 June to 23 July) and August (24 July to 25 August) moderate

*Figure 4 continued on next page*

*Figure 4 continued*

cooling (8–13°C) on bud set date for early- (green) and late- (blue) leafing trees (see **Figure 4—figure supplement 3** for extreme cooling effects). Analyses show effect size means ± 95% confidence intervals from linear models, including treatment and bud-type (apical vs lateral) as predictors. Early-leafing effects are calculated against the early-leafing control and late-leafing effects are calculated against the late-leafing control. The bud type effect is not shown. Number labels (1-4) above each point are shown to aid comparison between points 1–4 in the conceptual model (**Figure 1b**) and the observed effects. Positive values indicate advances in bud set and negative values indicate delays. All points are significantly different from zero (*p*<0.01) except point 1 (*p*=0.29).

The online version of this article includes the following figure supplement(s) for figure 4:

**Figure supplement 1.** Seasonal soil water content (%) for each treatment (experiment 1).

**Figure supplement 2.** Effects of slow spring development on the timing of autumn bud set in *Fagus sylvatica* (experiment 1).

**Figure supplement 3.** Effects of early-season development and extreme late-season cooling on the timing of autumn primary growth cessation in *Fagus sylvatica* (experiment 1).

**Figure supplement 4.** Effects of early-season development and late-season cooling on total bud growth in *Fagus sylvatica* (experiment 1).

**Figure supplement 5.** Effects of early-season development and late-season cooling on relative bud growth in *Fagus sylvatica* (experiment 1).

**Figure supplement 6.** Relationship between end of season bud length and the timing of primary growth cessation in *Fagus sylvatica* (experiment 1).

**Figure supplement 7.** Effects of early-season development and late-season cooling on the timing of leaf senescence in *Fagus sylvatica* (experiment 1).

---

(*p*=0.01). Conversely, post-solstice nighttime cooling delayed bud set by 3.8±3.5 days (*p*=0.03). Across all treatments, lateral buds set considerably earlier than apical buds (9.4±2.08 days, *p*<0.01). Treatment effects on relative bud growth can be found in **Figure 5—figure supplement 2**.

## Comparison of phenological metrics

Analyses using leaf senescence (50% drop in chlorophyll content) as the phenological marker produced results that were overall consistent with the bud set analyses. For example, in experiment 1, senescence was delayed by 4.22±1.61 days in late-leafing trees, and each day delay in spring leaf-out led to a 0.22±0.08 day delay in leaf senescence, closely matching with bud set. Across both experiments, the observed patterns were largely the same, although the strengths of each effect differed slightly, and overall, the effects on leaf senescence tended to be weaker (**Figure 4—figure supplement 7** and **Figure 5—figure supplement 3**). Out of all treatments, opposing responses from bud set and leaf senescence timing were only observed in the early-leafing extreme August cooling group (Treatment 9, bud effect = 1.9±2.59 days advancement, *p*=0.14, leaf effect = 1.77±3.53 days delay, *p*=0.32) from experiment 1, and the post-full-day cooling group from experiment 2 (bud effect = 5.2±1.81 days advancement, *p*<0.01, leaf effect = 1.66±3.34 days delay, *p*=0.32). Out of 66 pairwise comparisons of estimated marginal means (every treatment compared against every other treatment in the same experiment), 55 (83%) showed directional agreement between bud set and leaf senescence effects (**Figure 5—figure supplement 4**, adjusted $R^2$=0.49).

## Discussion
### Experimental forcing and scope of inference

The temperature manipulations applied in this study were intentionally designed as physiological forcing treatments rather than as simulations of realistic climatic scenarios. By imposing strong and temporally discrete constraints on development and sensitivity to cooling, we tested whether these physiological processes could be forced to advance or delay the timing of either autumn phenology or the compensatory point, as predicted by our conceptual model (**Figure 1**). Importantly, the exact date upon which the compensatory point was crossed was not (and with current knowledge cannot be) directly measured; instead, its position was inferred from changes in the direction and magnitude of temperature effects on autumn phenology before the solstice (June), directly after it (July), or later

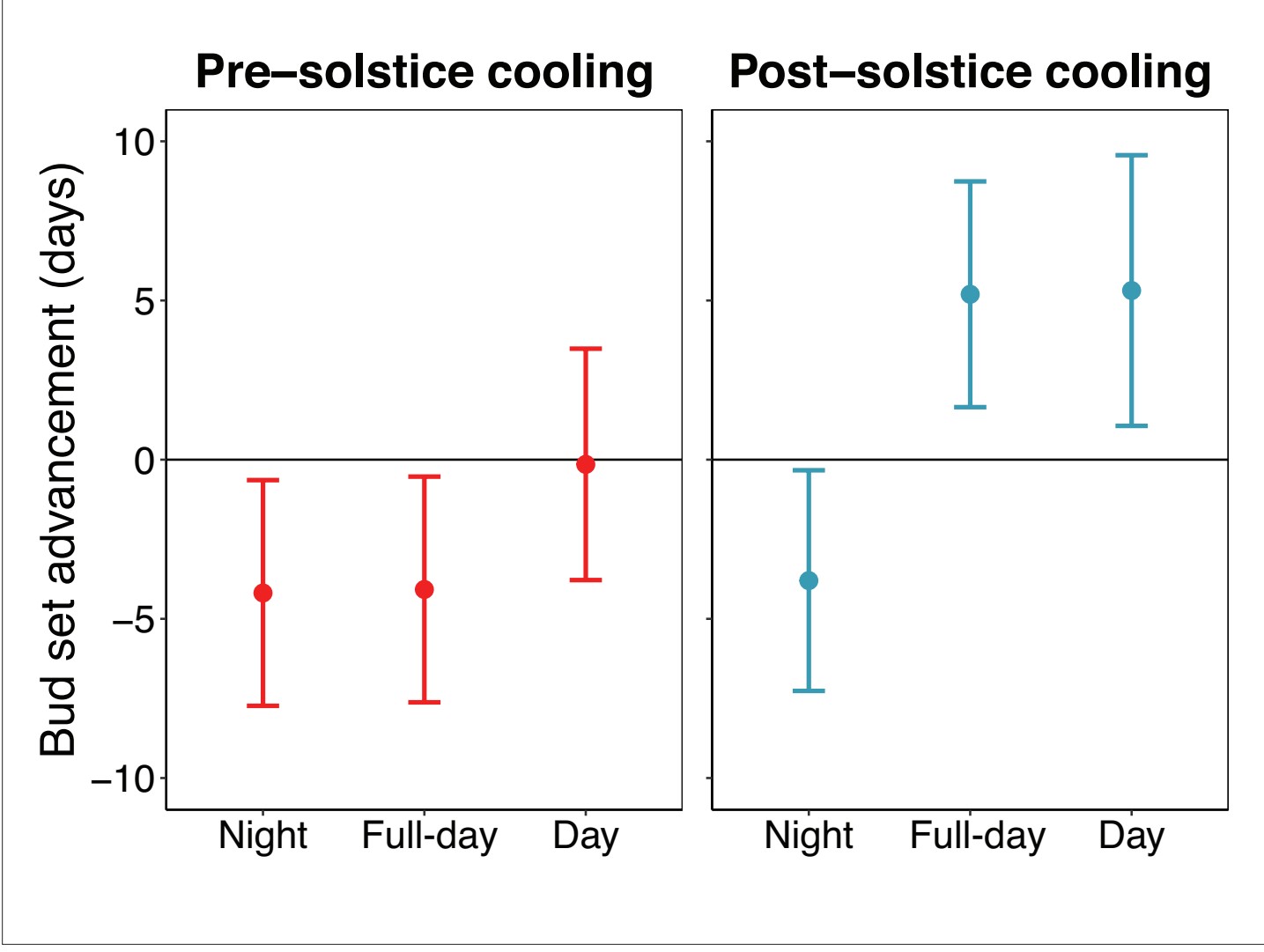

**Figure 5.** The pre- and post-solstice effects of night, full-day, and day cooling on the timing of autumn primary growth cessation in *Fagus sylvatica* (experiment 2). Effects of pre-solstice (22 May to 21 June) and post-solstice (22 June to 21 July) cooling on bud set date. Full-day cooling trees were continuously cooled to 8 °C, day cooling trees were cooled to 8 °C in the day and kept at 20 °C at night, night cooling trees were cooled to 8 °C at night and kept at 20 °C in the day. Analyses show effect size means ± 95% confidence intervals from linear models, including treatment and bud-type (apical vs lateral) as predictors. The bud type effect is not shown. All points are significantly different from zero (*p*<0.04) except pre-solstice day cooling (*p*=0.94).

The online version of this article includes the following figure supplement(s) for figure 5:

**Figure supplement 1.** Bud set timing (day-of-year) for each control treatment (experiment 2).

**Figure supplement 2.** Effects of pre- and post-solstice daytime, nighttime, and full-day cooling on relative bud growth in *Fagus sylvatica* (experiment 2).

**Figure supplement 3.** The pre- and post-solstice effects of night, full-day, and day cooling on the timing of autumn leaf senescence in *Fagus sylvatica* (experiment 2).

**Figure supplement 4.** Treatment effects on the timing of bud set (x-axis) and leaf senescence (y-axis) across our two experiments.

**Figure supplement 5.** Photosynthesis rates of *Fagus sylvatica* trees during the pre-solstice treatment window (experiment 2).

in the growing season (August). Accordingly, the compensatory point should be interpreted as an inferred conceptual node that structures seasonal temperature responses, rather than as a directly observed phenological event. Given the large individual variation expected in phenological experiments, we used single experimental populations of single provenance beech saplings to minimise uncontrolled variation arising from genetic differences (*Meger et al., 2021*). The primary inference

of these experiments concerns causality and constraint, i.e., whether these processes are capable of producing the observed patterns, rather than quantitative predictions of ecosystem responses to contemporary or future climates.

Our experiments on European beech test how bud set—a marker of primary growth cessation—and leaf senescence respond to monthly and diel temperature changes around the summer solstice. We found large differences in the responses between trees subjected to daytime versus nighttime cooling before and after the solstice (*Figure 5*) and between early- and late-leafing trees (*Figure 4*). These differences suggest that the ESD effect not only influences the timing of autumn phenology per se but also plays a critical role in determining the actualised timing of the reversal of phenological responses to temperature after the summer solstice. Therefore, our findings support our conceptual model (*Figure 1*), highlighting development as a key factor underpinning the Solstice-as-Phenology-Switch hypothesis. In the following sections, we discuss the effects of pre-solstice and post-solstice air temperature on development, along with the distinct impacts of daytime versus nighttime temperature variations.

## Early-season development rates alter responses to late-season temperature

Our trans-solstice climate manipulation experiments showed that trees delayed the end of their growing season in response to slow/late early-season development (delayed spring leaf-out; ESD effect). Regardless of the summer cooling treatments, late-leafing trees consistently set buds and senesced leaves after their early-leafing counterparts (*Figure 4—figure supplement 2*), with each day of delay in spring leaf-out delaying bud set by an average of 0.24 days and senescence by 0.22 days. This is in line with previous studies demonstrating a tight linkage between within-year variations in spring and autumn phenology (*Fu et al., 2014*; *Keenan and Richardson, 2015*; *Signarbieux et al., 2017*), likely governed by developmental constraints (*Zohner et al., 2023*), buildup of water and nutrient stress (*Bigler and Vitasse, 2021*; *Buermann et al., 2018*; *Paul and Foyer, 2001*), or leaf aging (*Lim et al., 2007*).

Whilst lateral buds tended to set earlier than apical buds, the effect was not consistent across leafing groups, suggesting that bud position interacts with developmental context rather than exerting a uniform effect. This inconsistency may reflect differences in non-structural carbohydrate distribution and hormonal gradients among individuals at different developmental stages (*Barbier et al., 2017*; *Powell, 1988*; *Singh et al., 2022*). Under reduced early-season development, earlier lateral compared to apical bud set suggests enhanced apical dominance with increased allocation of resources to the sapling's primary shoot.

Summer cooling treatment [LST effect] explained more variance in bud set timing than leaf-out day-of-year [ESD effect] (Adjusted $R^2$=0.15 and 0.09, respectively), with a clear interaction between the two effects. July cooling induced a delay in bud set dates 3.4 times greater in late-leafing trees compared to early-leafing ones (4.8 vs 1.4 days delay), which agrees with the expectations derived from our conceptual model (*Figure 1*). This dependence of autumn phenological responses to summer cooling on developmental progress demonstrates flexibility in the compensatory point between the antagonistic influences of the ESD and LST effects. Although we did not directly identify a discrete timing at which this compensatory point occurred, the pronounced difference in July treatment effects between early- and late-leafing trees suggests a shift in the seasonal window during which cooling delays, rather than advances, autumn phenology. Together, these results support a development-dependent flexibility in the effective timing of the reversal in temperature responses, consistent with a flexible compensatory point governed by developmental progression.

In natural systems, these responses may arise from interannual variation in early-season development, leading to year-to-year differences in the timing at which cool temperatures switch from delaying autumn phenology to advancing it. This would offer a physiological explanation for the advancement in the response reversal observed between 1966 and 2015 in *Fagus sylvatica*, *Aesculus hippocastanum*, *Quercus robur* and *Betula pendula* (*Zohner et al., 2023*). However, extrapolations to more complex natural ecosystems should be made with caution as our experimental design prioritised mechanistic inference over generalisability and predictive power. In line with this focus, summer cooling treatment (LST effect), leaf-out date (ESD effect), and bud type together explained a substantial, though not majority, proportion of the variation in bud set timing (24%). Several physiological mechanisms could,

in principle, contribute to this unexplained variation. For example, declining photosynthetic assimilation has been linked to accelerated senescence (*Krieger-Liszkay et al., 2019*). However, experimental reductions of August photosynthetic rates in beech (52–72%), achieved through either cooling or shading, demonstrated that photosynthetic assimilation was not a driver of phenological responses in beech (*Zohner et al., 2023*). This suggests that photosynthesis is unlikely to explain the residual variation observed here.

Rather than missing explanatory variables, much of the remaining variation is attributable to individual-level differences, as suggested by the relatively low intra-class correlation coefficient [0.26] (*Nakagawa et al., 2026*). Individual variation likely reflects genetic diversity—even within a single provenance population—and epigenetic mechanisms that influence phenological timing (*Carneros et al., 2017*; *Crawley and Akhteruzzaman, 1988*; *Malyshev et al., 2022*; *Scotti et al., 2016*; *Solvin and Steffenrem, 2019*). Additional sources of within-treatment variability may include measurement error, microclimatic heterogeneity, and baseline differences among individuals, although steps were taken to minimise these throughout the experiment. However, because control trees did not experience chamber time, we cannot entirely rule out some influence of this factor on our comparisons. In this context of high individual variability, leaf-out timing (ESD effect) and summer cooling treatment (LST effect) together explaining 23.4% of variation in bud set timing demonstrates the mechanistic importance of these processes.

August cooling induced comparable advances in bud set timing in both early- and late-leafing trees (4.4–4.5 days). The diminishing difference in cooling responses between the two groups from July to August suggests that, by August, phenology is primarily governed by the LST effect (see direct temperature effect in *Figure 1b*). The observed increase in the LST effect from July to August may be driven by a reduction in the influence of the ESD effect or by declining daylength (*Kramer, 1936*; *Petterle et al., 2013*; *Singh et al., 2021*), but more likely by an interaction between the two. Firstly, a weakening in the ESD effect is expected as individuals approach completion of their developmental requirements. Then, as days shorten, trees become increasingly responsive to cooling (*Delpierre et al., 2009*; *Körner et al., 2016*), strengthening the LST effect. Together, these changes can explain why, by August, trees responded similarly to cooling regardless of their earlier developmental trajectories (*Figure 1*). These findings highlight the critical interaction between photoperiod and temperature in shaping the autumn phenology of *Fagus sylvatica*.

## Effects of daytime vs. nighttime temperature

In the second experiment, we independently manipulated daytime and nighttime temperatures. The imposed decoupling of daytime and nighttime temperatures was not intended to represent common meteorological conditions. Rather, it was used to isolate processes that are inherently diel in nature. In trees, cell division and expansion predominantly occur at night (*Steppe et al., 2015*; *Mencuccini et al., 2017*; *Zweifel et al., 2021*), whereas carbon assimilation occurs primarily during the day (*Figure 6a*). By experimentally separating these phases, we tested whether phenological responses depend on when within the diel cycle temperature constraints are applied. The strong and opposing responses observed under these forced asymmetries indicate that diel timing is a critical axis of physiological control, even if natural temperature cycles rarely produce such extreme contrasts.

Our experiments showed that daytime and nighttime temperatures had different effects on autumn phenology before and after the summer solstice: before the solstice, daytime cooling had no impact on the timing of bud set, while nighttime cooling and full-day cooling delayed bud set by 4.2 and 4.1 days, respectively (*Figure 5*). This pattern was similar when using leaf senescence as the end-of-season marker. Because experimental cooling reduced photosynthesis by 69–83% across all treatments (*Figure 5—figure supplement 5*), these effects are unlikely to be explained by changes in carbon assimilation. Instead, they highlight the importance of developmental processes—such as cell division and expansion—which mostly occur at night (*Steppe et al., 2015*; *Mencuccini et al., 2017*; *Zweifel et al., 2021*; *Figure 6a*). The developmental arrest induced by night cooling—and, therefore, also by full-day cooling—likely slowed growth and maturation, effectively extending the growing season (ESD effect).

After the solstice, daytime and nighttime cooling of 8 °C elicited opposite responses in both phenological markers, with weaker effects on senescence. Trees subjected to post-solstice daytime cooling and full-day cooling set their buds earliest, on average, more than 5 days earlier than controls. As days

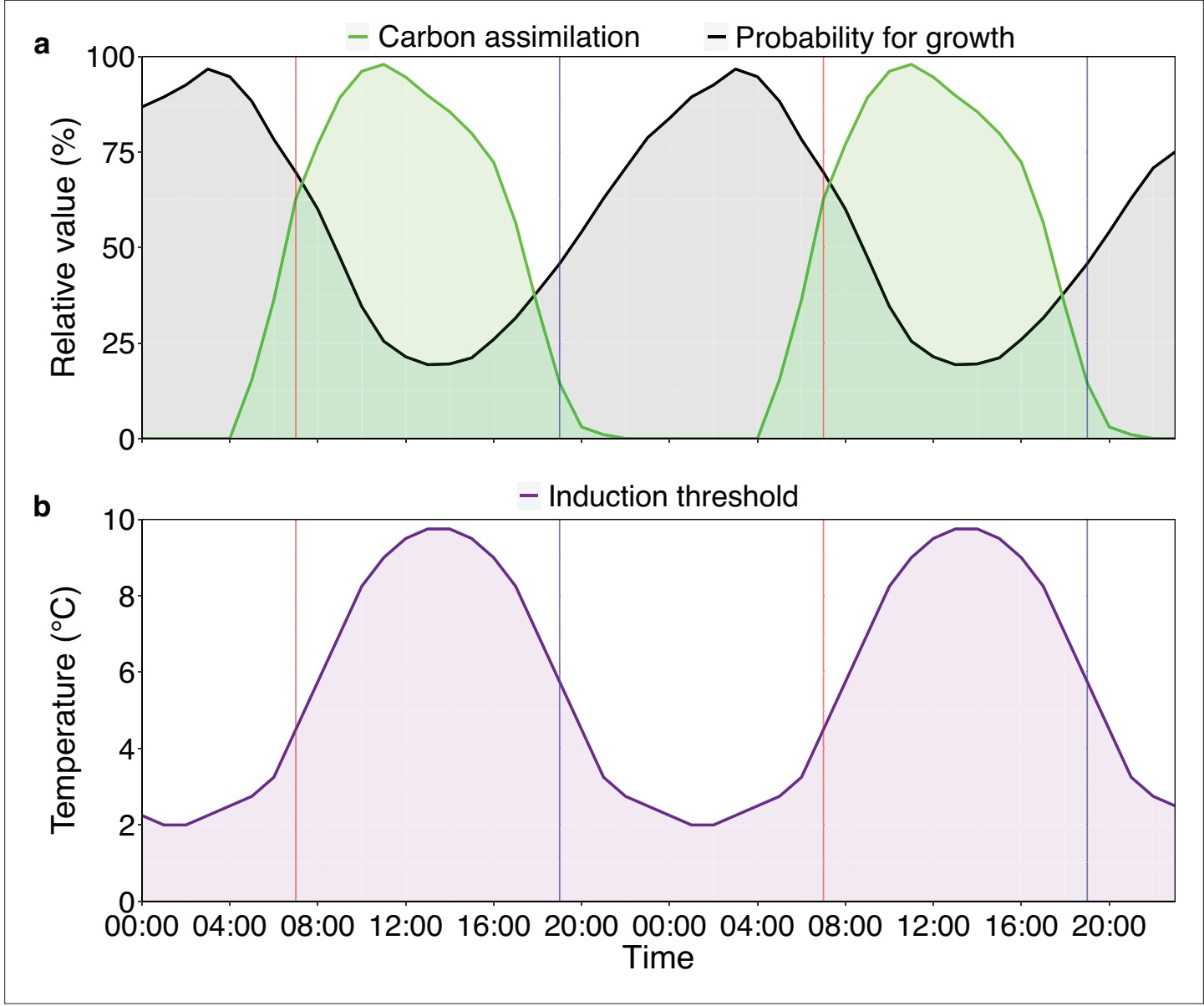

**Figure 6.** Diel patterns of relative growth, photosynthetic rate, and theoretical thresholds for cold-induced bud set in *Fagus sylvatica*. Vertical red lines indicate the start of the day, and vertical blue lines indicate the start of the night, marking the boundaries between the 12 hr treatment windows used in experiment 2. (**a**) The green curve shows relative carbon assimilation rate, the raw values were taken from the literature in a study that measured assimilation rates under controlled conditions (**Urban et al., 2014**). Values were linearly interpolated between measurements, then converted to a percentage of the peak value. Finally, the curve was smoothed by taking the running mean of the target value, the previous value, and the following value. The black curve shows the relative probability for growth; the raw values were taken from the literature in a study that measured growth rates in the field (**Zweifel et al., 2021**), then processed in the same way as the assimilation data. At night, low temperatures slow the trees' developmental processes, possibly leading to delayed tissue maturation, while low temperatures during the day reduce photosynthetic activity. (**b**) The Diel Cooling Hypothesis: As days shorten after the solstice, autumn bud set becomes increasingly responsive to cooling. Temperatures below a certain threshold induce overwintering responses, advancing autumn phenology. Our results indicate that daytime cooling of 8 °C is below this threshold. However, because daily temperatures reach their minimum during the night, trees' induction thresholds should be lower at night than during the day. Post-solstice night-time cooling of 8 °C may thus have delayed bud set by slowing development rather than inducing overwintering responses.

The online version of this article includes the following figure supplement(s) for figure 6:

**Figure supplement 1.** The Diel Cooling Hypothesis: Conceptual model of pre- and post-solstice effects of day and nighttime temperatures on the autumn phenology of *Fagus sylvatica*.

shorten, the temperature threshold inducing this growth cessation must be lower at night, otherwise, trees would risk prematurely ending their growing season due to the daily minimum temperatures occurring at night (*Figure 6b*). This may explain why post-solstice nighttime cooling delayed, rather than advanced, autumn phenology: cooling of 8 °C was sufficient to trigger overwintering responses—growth cessation and maturation of perennating tissues—when applied during the day but insufficient when applied at night (see *Figure 6—figure supplement 1* for a conceptual schematic).

An alternative explanation is that daytime cooling indirectly triggered dormancy induction by suppressing photosynthesis (*Figure 6a*), under the assumption that trees prioritise dormancy once photosynthesis is no longer possible (*Körner et al., 2016*). Though, as discussed above, late-season photosynthesis appears to assert little control over the autumn phenology of *F. sylvatica* (*Zohner et al., 2023*). Taken together, these results support a Diel Cooling Hypothesis, in which the impact of cooling depends on whether it occurs during the day or at night, because the temperature thresholds that trigger growth cessation differ between daytime and nighttime conditions.

Nighttime cooling always led to a delay in bud set of approximately four days (*Figure 5*). As trees mainly grow at night (*Mencuccini et al., 2017*; *Steppe et al., 2015*; *Zweifel et al., 2021*), colder nighttime temperatures likely slowed down key developmental processes, such as meristematic activity, tissue expansion, and maturation, which in turn delayed primary growth cessation (*Figure 6*). Similar responses may occur in other temperate tree species: cold autumn nights have been shown to delay growth cessation and slow bud development in *Populus, Pinus,* and *Picea* species (*Kalcsits et al., 2009*; *Kramer, 1957*; *Malcolm and Pymar, 1975*). Moreover, the reversal in responses to temperature after the summer solstice has been observed consistently across Northern Hemisphere temperate forests (*Zohner et al., 2023*). Further experiments should explicitly test these diel responses in other temperate tree species as well as other provenances of beech.

## Support for the Solstice-as-Phenology switch hypothesis

The Solstice-as-Phenology-Switch hypothesis posits that temperature effects on autumn phenology reverse around the summer solstice, but does not assume that this reversal occurs exactly on June 21 (*Zohner et al., 2023*). Instead, the solstice represents the earliest point at which such a reversal can emerge, marking the onset of declining daylengths that enable phenological sensitivity to late-season cooling. Here, we show how the antagonistic influences of the ESD effect and the LST effect can jointly determine the timing of this reversal.

Under this framework, the LST effect is 'switched-on' after the solstice as daylength begins to decline (*Figure 1*). However, the observed reversal in temperature responses occurs only once early-season developmental effects (ESD effect) are balanced by increasing sensitivity to cooling (LST effect)—that is, at a compensatory point. Therefore, the timing of this reversal is not fixed to a calendar date but varies with developmental progression. Our conceptual model (*Figure 1*) explicitly incorporates this flexibility, showing that the timing of the reversal depends on realised early-season development (ESD effect): under conditions that slow development (e.g. late leaf-out), the compensatory point is reached later in the season, whereas faster development advances it.

Our experiments support this framework: pre-solstice full-day cooling delayed bud set, whereas post-solstice full-day cooling advanced it, with differences between early- and late-developing individuals consistent with model predictions. Moreover, the contrasting impacts of daytime vs. nighttime cooling demonstrate that diel conditions further shape when the reversal is expressed. Thus, our findings support the Solstice-as-Phenology-Switch hypothesis and extend it by showing that its flexibility arises from interactions between developmental progression, diel temperature responses, and photoperiod.

Having established these effects under controlled physiological forcing, we next consider how the underlying mechanisms may operate under natural, fluctuating conditions. While beech trees are unlikely to experience sustained temperature regimes equivalent to the imposed chamber conditions, analogous developmental constraints can arise during cold springs, late cold spells following budburst, or at high-elevation and continental sites where temperatures remain low despite increasing photoperiod. Moreover, because developmental progression integrates temperature cumulatively over time, even short episodes of strong cooling can exert lasting carry-over effects on seasonal timing (*Chuine, 2000*; *Liu et al., 2024*). The relevance of our findings, therefore, lies not in the specific

temperature regimes applied, but in identifying the processes through which temperature and development interact to shape autumn phenology.

Our results demonstrate that accelerating developmental progression—particularly via nighttime temperature effects—can advance the timing at which cooling begins to trigger growth cessation and senescence in European beech. This provides a mechanistic framework for interpreting phenological responses under climate warming, rather than making direct quantitative predictions under specific climate scenarios. These findings highlight the importance of accounting for developmental context and diel temperature sensitivity in phenological models of temperate trees.

## Acknowledgements

CMZ was supported by the SNF Ambizione Fellowship programme (no. PZ00P3_193646) and TWC by DOB Ecology and the Bernina Foundation.

## Additional information

### Funding

| Funder | Grant reference number | Author |
|---|---|---|
| DOB Ecology | | Thomas W Crowther |
| Bernina Foundation | | Thomas W Crowther |
| Schweizerischer Nationalfonds zur Förderung der Wissenschaftlichen Forschung | PZ00P3_193646 | Constantin M Zohner |

The funders had no role in study design, data collection and interpretation, or the decision to submit the work for publication.

### Author contributions

Dominic Rebindaine, Conceptualization, Data curation, Formal analysis, Investigation, Visualization, Methodology, Writing – original draft, Project administration, Writing – review and editing; Thomas W Crowther, Supervision, Funding acquisition, Writing – review and editing; Susanne S Renner, Writing – review and editing; Zhaofei Wu, Yibiao Zou, Lidong Mo, Haozhi Ma, Raymo Bucher, Conceptualization, Investigation, Writing – review and editing; Constantin M Zohner, Conceptualization, Formal analysis, Supervision, Funding acquisition, Investigation, Methodology, Writing – review and editing

### Author ORCIDs

Dominic Rebindaine  https://orcid.org/0000-0002-8110-2417
Thomas W Crowther  https://orcid.org/0000-0001-5674-8913
Susanne S Renner  https://orcid.org/0000-0003-3704-0703
Zhaofei Wu  https://orcid.org/0000-0001-6333-118X
Yibiao Zou  https://orcid.org/0000-0002-4741-0934
Haozhi Ma  https://orcid.org/0000-0003-0709-1438
Raymo Bucher  https://orcid.org/0000-0001-7607-0476
Constantin M Zohner  https://orcid.org/0000-0002-8302-4854

Reviewer #1 (Public review): https://doi.org/10.7554/eLife.107554.4.sa1
Reviewer #2 (Public review): https://doi.org/10.7554/eLife.107554.4.sa2
Author response https://doi.org/10.7554/eLife.107554.4.sa3

## Additional files

### Supplementary files

Supplementary file 1. Average phenology and bud growth by treatment for experiment 1. Mean and

95% confidence intervals of 50% leaf-out (day-of-year BBCH15), bud set (day-of-year), leaf-off (day-of-year 50% loss of leaf chlorophyll), and absolute bud growth (cm) for each treatment in experiment 1.

Supplementary file 2. Average autumn phenology and bud growth by treatment for experiment 2. Mean and 95% confidence intervals of bud set (day-of-year), leaf-off (day-of-year 50% loss of leaf chlorophyll), and absolute bud growth (cm) for each treatment in experiment 2.

MDAR checklist

## Data availability

The code and data for this study are available at Zenodo: https://zenodo.org/doi/10.5281/zenodo. 19368812 (*Rebindaine et al., 2026*).

The following dataset was generated:

| Author(s) | Year | Dataset title | Dataset URL | Database and Identifier |
|---|---|---|---|---|
| Rebindaine D | 2026 | Developmental constraints mediate the reversal of temperature effects on the autumn phenology of European beech after the summer solstice | https://doi.org/ 10.5281/zenodo. 19368812 | Zenodo, 10.5281/ zenodo.19368812 |

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
