## [Editor Report · eLife Assessment]

This article presents **valuable** findings on how the timing of cooling affects autumn bud set in European beech saplings. The study leverages extensive experimental data and provides an interesting conceptual framework for the various ways in which warming can affect bud set timing. The statistical analysis is very well considered, while indicating some factors that may temper the authors' claims. The factorial experiments offer **solid** support.

---

## [Referee Report · Reviewer #1 (Public review)]

[Editors' note: this version has been assessed by the Reviewing Editor without further input from the original reviewers. The authors have addressed the comments raised in the previous round of review.]

Summary:

This study provided key experimental evidence for the "Solstice-as-Phenology-Switch Hypothesis" through two temperature manipulation experiments.

Strengths:

The research is data-rich, particularly in exploring the effects of pre- and post-solstice cooling, as well as daytime versus nighttime cooling, on bud set timing, showcasing significant innovation. The article is well-written, logically clear, and is likely to attract a wide readership.

---

## [Referee Report · Reviewer #2 (Public review)]

In 'Developmental constraints mediate the reversal of temperature effects on the autumn phenology of European beech after the summer solstice', Rebindaine and co-authors report on two experiments on Fagus sylvatica where they manipulated temperatures of saplings between day and night and at different times of year. I think the experiments are interesting, but note that the treatments are extreme compared to natural conditions. Further, given that much of the experiment happened outside, I am not sure how much we can generalize from one year for each experiment, especially when conducted on one population of one species.

---

## [Author Response]

The following is the authors’ response to the previous reviews

**eLife Assessment**
This article presents useful findings on how the timing of cooling affects the timing of autumn bud set in European beech saplings. The study leverages extensive experimental data and provides an interesting conceptual framework for the various ways in which warming can affect but set timing. The statistical analysis is compelling, but indicates some factors that may temper the authors' claims, while the designs of experiments offer incomplete support for the current claims as they rely on one population under extreme conditions for only one year each while a confounding effect (time in a chamber) sometimes lacks a control.

We thank the editor and reviewers for their consideration of our revised manuscript and for their constructive suggestions. In response to the editor’s guidance, we have ensured that: (1) the experimental design is clearly presented as physiological forcing, (2) the Solstice-as-Phenology-Switch concept is explicitly defined, limited, and framed as inferred, (3) conclusions are strictly aligned with the scope of the evidence, and limitations are acknowledged transparently.

We hope these revisions fully address the remaining concerns and clarify both the conceptual framework and the appropriate scope of inference.

**Public Review:**

**Reviewer #1 (Public review):**
The authors identified the summer solstice (June 21) as a phenological "switch point", but the flexibility of this switch point remains poorly understood. A more precise explanation of what "flexibility" means in this context is needed, along with a description of the specific experimental results that would demonstrate this flexibility.

We agree that the concept of “flexibility” required clearer definition and a more explicit link to the experimental results. In the Introduction, we now explicitly define flexibility as the capacity for the effective timing of the phenological switch to shift earlier or later depending on developmental progression, rather than occurring at a fixed calendar date. This switch occurs at the compensatory point between the antagonistic influences of early-season development [ESD effect] and late-season temperature [LST effect](L92-98). We have extended and clarified our explanation of the summer solstice’s role in this framework (L69-90). We propose that the solstice acts as an environmental switch that initiates the LST effect, as declining daylengths signal trees to become responsive to late-season cooling (L92-94). The compensatory point then occurs where the advancing ESD effect is balanced by the delaying LST effect. This point should therefore not be fixed to a calendar date but instead vary with developmental progression each year (L75-95).

In the Discussion, we clarify that flexibility is demonstrated experimentally by the observation that the magnitude of July cooling effects (LST effect) on autumn phenology depend on prior developmental rate (ESD effect) [3.4 times greater delay in late-leafing trees], indicating that the position of the compensatory point is development-dependent rather than fixed to June 21 (L398-410). We have made consistent edits throughout the Discussion, in particular in the ‘Support for the Solstice-as-Phenology-Switch Hypothesis’ subsection (L514-530).

The experiment did not directly measure the specific date of the phenological switch point. Instead, it was inferred by comparing temperature effects before and after the solstice. The manuscript should clearly state that this switch point remains an inferred conceptual node rather than a directly measured variable.

We fully agree and have clarified this in the revised manuscript. In the Discussion, we now clearly state that the compensatory point is a conceptual node inferred from responses to cooling before the solstice (June), directly after it (July), or later in the growing season (August) rather than a directly observed phenological event (L352-358 & L405-406).

In Experiment 1, the effect of bud type (terminal vs. lateral) was inconsistent across the overall model and the different leafing groups. The authors should provide a more thorough discussion of potential reasons for this inconsistency.

This inconsistency reflects biological complexity. In the Discussion, we now expand our interpretation to note that terminal and lateral buds may differ in developmental status, resource allocation and hormonal context. We emphasize that bud-type effects are therefore expected to be context-dependent and to interact with wholeplant developmental state, which plausibly explains why effects differ across leafing groups and models (L390-396).

In addition, the statistical model for Experiment 1 indicates that the measured variables (summer cooling and leaf emergence date) explain only 23.4% of the variation in bud formation timing. This leaves over 76% of the variation unexplained, suggesting that other important factors are involved. The discussion should address this limitation in greater depth, moving beyond a focus on the measured variables.

We now discuss the explained and unexplained variance in more detail. We also make it clear that our experiment was designed to test specific mechanistic pathways rather than to fully explain all phenological variability or maximise predictive power (L417-419).

In the Discussion, we acknowledge that a substantial fraction of variation remains unexplained (L419-421). We discuss the possibility of other physiological mechanisms, such as photosynthetic assimilation, contributing to the unexplained variation (L421-427). However, large inter-individual variability is commonplace in autumn phenology. A low intra-class correlation coefficient (ICC = 0.26; see L276-280 for methods) suggests much of the remaining variation is attributable to individual-level differences rather than missing explanatory variables (L429-431). In line with the literature, we suggest that genetic and epigenetic differences likely contributed significantly to inter-individual variation, even within a single provenance population (L431-434). In this context of high individual variability, leaf-out timing (ESD effect) and summer cooling treatment (LST effect) together explaining 23.4% of variation in bud set timing is biologically meaningful and demonstrates the mechanistic importance of these processes (L438-441). For completeness, we also briefly discuss alternate sources of within-treatment variability (L434-437).

**Reviewer #2 (Public review):**
I think the experiments are interesting, but I found the exact methods of them somewhat extreme compared to how the authors present them.

We appreciate this concern and have substantially revised the manuscript to clarify the experimental logic. In the Introduction, we now state explicitly that the study uses temperature regimes that were designed as strong physiological forcing treatments, intended to deeply constrain development and isolate mechanisms rather than to simulate natural or future climatic conditions (L113-115).

In the Methods, we have enhanced our description of the non-linear effects of temperatures below 10°C on physiological processes (L154-158).

At the start of the Discussion, we have added a dedicated paragraph clarifying the scope of inference: the experiment tests causality and constraint (i.e. whether specific physiological processes can drive phenological shifts), not quantitative responses under realistic climate scenarios (L346-363). Throughout the Discussion, we have revised language that could be read as scenario-based interpretation, replacing it with mechanistic phrasing.

Further, given that much of the experiment happened outside, I am not sure how much we can generalize from one year for each experiment, especially when conducted on one population of one species.

Given the large individual variation expected in phenological experiments, we used single experimental populations of single provenance beech saplings to minimise uncontrolled for variation arising from genetic differences (L358-360). This allowed us to elucidate mechanisms despite noisy biological heterogeneity associated with phenology.

In the last round of revision, we toned down statements of generalisation. In the Discussion, we now go further to clarify what mechanistic understanding can be gleamed directly from our findings and then cautiously make suggestions how these mechanisms may play out in natural systems. We repeatedly state the intention of the study as mechanistic inference rather than predictive power, e.g. “However, extrapolations to more complex natural ecosystems should be made with caution as our experimental design prioritised mechanistic inference over generalisability and predictive power.” (L417-419). Alongside our previous calls for tests on other species, we now additionally call for tests on other provenances of beech (L511-512).

I was also very concerned by the revisions.

If this concern stems from the confusion regarding line-numbers and the two submitted versions of the manuscript (with tracked changes and without tracked changes; as required by eLife), then we hope that situation is now clarified. Otherwise, the authors do not understand why our previous revisions would be perceived as being concerning. Regardless, we have made every attempt to address the remaining comments comprehensively.

Further, I am at a loss about their hypothesis, when they write in their letter: "Importantly, the Solstice-asPhenology-Switch hypothesis does not assume that the reversal is fixed to June 21." Why on earth reference the solstice if the authors do not mean to exactly reference the solstice?

We appreciate this important conceptual point. The Solstice-as-Phenology-Switch hypothesis is central to our conceptual model and therefore requires clear explanation. In concert with our changes in response to Reviewer 1’s comment regarding flexibility, we have substantially revised and improved our description of this hypothesis (L69-108).

Whilst the summer solstice is fixed to a calendar date (June 21), the timing of when trees change their autumn phenological responses to temperature is not (L88-90 & L515-517). This occurs when the compensatory point of two antagonistic effects is crossed. Higher early-season development rates (which are driven by temperature) have an advancing (negative) effect on autumn phenology, which we now refer to as the ESD effect (L71-78). Warmer late-season temperatures have a delaying (positive) effect because trees become phenologically susceptible to cooling, i.e. overwintering responses are induced in response to cooling, which we now refer to as the LST effect (L78-82). The point in time when these two effects balance each other out, i.e. the net effect = 0, is the compensatory point (L95-97 & L523-525). The reason this point occurs after the solstice, is because the LST effect only becomes active when days begin to shorten (L92-94 & L522-523). The solstice acts as an environmental switch, initiating trees’ susceptibility to cooling. Therefore, the solstice is referenced in the hypothesis because it forms a daylength barrier. In this framework, the compensatory point cannot occur earlier than the solstice because day lengths are still increasing (L517-519).

In the Introduction and Discussion, we clarify that the solstice is referenced as a biologically meaningful photoperiodic cue, not as a fixed threshold date. We now emphasise that the hypothesis concerns a seasonal reversal in responses to temperature structured around photoperiod, whose effective timing depends on developmental state, rather than a reversal occurring precisely on June 21. To avoid confusion, we have reworded phrases such as “summer solstice effect reversal” to “reversal of phenological responses to temperature after the summer solstice” (L371). In accordance, we have also changed the title to “Developmental constraints mediate the reversal of temperature effects on the autumn phenology of European beech after the summer solstice”.

The following comments stem from the first round of review. We have previously revised the manuscript in accordance with these comments. For most of these points we do not see further cause for changes except for any overlap with comments above. We therefore predominantly copy our previous responses in quotes for clarity, the exception being the comment regarding the framing of our results in relation to natural systems.

The comments below relate to my original review with many of them still applying.Methods: As I read the Results I was surprised the authors did not give more info on the methods here. For example, they refer to the 'effect of July cooling' but never say what the cooling was. Once I read the methods I feared they were burying this as the methods feel quite extreme given the framing of the paper.

“We understand the concern regarding the structure of the manuscript and note that the methods section was moved to the end of the paper in accordance with eLife’s recommended formatting. We have now moved the methods section before the results to ensure that readers are familiar with the treatments before encountering the outcomes.

Regarding presentation, treatment details are now described in both the Methods and the relevant figure legends. Given this structure, we have chosen not to restate the full treatment conditions in the main Results text to avoid repetition.”

The paper is framed as explaining observational results of natural systems, but the treatments are not natural for any system in Europe of which I have worked in. For example a low of 2 deg C at night and 7 deg C during the day through end of May and then 7/13 deg C in July is extreme. I think these methods need to be clearly laid out for the reader so they can judge what to make of the experiment before they see the results.

We appreciate the reviewer’s concern regarding the use of relatively extreme temperature treatments and the need to ensure that our conclusions are consistent with the motivation for using them. The manuscript was also revised in this regard in the previous round, and we copy the relevant responses at the bottom of this response. Despite this, we agree that further explanation of how our experimental treatments suited the aims of our study was still required.

The aim of these treatments was not to reproduce typical ambient conditions, but to act as a mechanistic probe. Such mechanisms are not readily identifiable from observations or mild manipulations, because the expected effects are small relative to natural variability; stronger perturbations are therefore required to generate a diagnostic contrast. By strongly constraining development in the early-season, and by providing a robust cooling signal in the late-season, we sought to reveal the causal structure underlying the observed solstice-related reversal in temperature effects on autumn phenology.

Temperatures below 10°C intensively slow down cell division and mitotic rates, these rates then rapidly and non-linearly approach 0 as temperatures drop towards 0°C (Körner, 2021). As reflected in L152-158 of the revised manuscript, we selected a spring cooling regime of 2–7 °C to strongly slow developmental processes while maintaining a clear thermal safety margin that eliminates the risk of frost damage. Although a milder cooling regime (e.g. 5–10 °C) would be less extreme, it would also be expected to produce only a comparatively small reduction in developmental rates, thereby substantially reducing our ability to generate distinct early- and late-developing individuals and to detect carry-over effects on autumn phenology. Applying strong cooling therefore increases signal-to-noise and allows us to detect the underlying mechanism, which would not be possible with temperature treatments that represent average contemporary climatic variation.

The use of conditions out with the norm is a standard practice to elucidate mechanisms in ecology, where organisms are often pushed to their physiological limits or transplanted into environments fundamentally different to those which they are adapted (Somero, 2010; Berend et al., 2019). Experiments targeting autumn phenology have utilised a broad range of environmental conditions from moderate to extreme manipulations (Tanino et al., 2010). For example, to test the controls of growth cessation and dormancy induction in Prunus species, one study applied a range of treatments including constant 9°C temperature and 24 hour photoperiod between April and July (Heide, 2008).

Our experimental design aimed to reduce rates of development, cell division and maturation. In the Methods, we describe this aim and clearly state that the experimental design was not intended to mimic natural climatic variation (L154-156 & L181-186). Importantly, our conclusions are framed at the level of direction, timing, and interaction of effects, rather than the magnitude expected under contemporary or future field conditions (L360-363).

This framing intends to reflect the primary inference of this study, which concerns when and why temperature effects reverse around the solstice, and how this timing depends on developmental state and diel temperature exposure, rather than making quantitative predictions for present-day or future climates. This aligns our conclusions with the experimental design. We have further revised the Discussion to explain these aims and conclusions more clearly, including the addition of a subsection at the beginning titled “Experimental forcing and scope of inference” (L346-363). We have also set up this expectation in the Introduction (L113-115).

Additionally, we have improved the Discussion in a number of related aspects.

We explicitly separate mechanistic conclusions and any relation to natural systems, remaining cautious to not overgeneralise or overstate our findings (L417-419).

We now include a dedicated paragraph explaining that, although these specific conditions are not likely to be found in beech’s range, analogous developmental constraints can arise during cold springs, late cold spells following budburst, or at high-elevation and continental sites where temperatures remain low despite increasing photoperiod (L540-545, L583-588). We further explain that because developmental progression integrates temperature cumulatively over time, even short episodes of strong cooling can exert lasting carry-over effects on seasonal timing, thereby linking the forced experimental responses to processes relevant under natural, fluctuating conditions (L545-550).

We explicitly state that the decoupling of day and night temperatures was not intended to represent realistic meteorological states (L458-460). We explain that this design was used diagnostically to isolate inherently diel physiological processes (e.g. nocturnal growth, cell division and expansion versus daytime carbon assimilation), and that the observed responses demonstrate the importance of diel timing of temperature exposure rather than the realism of the imposed cycles (L460-468).

Previous response:

We recognise that our temperature treatments were severe and do not mimic real world scenarios. They were deliberately designed to create large contrasts in developmental rates, thereby maximising our ability to detect the mechanisms underpinning the solstice switch. For example, the severe cooling between 4 April and 24 May was specifically designed to slow spring development as much as possible without damaging the plants. We have added text in the Methods to clarify this aim.

I also think the control is confounded with growth chamber experience in Experiment 1. That is, the control plants never experience any time in a chamber, but all the treatments include significant time in a chamber. The authors mention how detrimental chamber time can be to saplings (indeed, they mention an aphid problem in experiment 2) so I think they need to be more upfront about this. The study is still very valuable, but -- again -- we may need to be more cautious in how much we infer from the results.

We appreciate the reviewer’s concern about the potential confounding effect of chamber exposure in experiment 1. We have now discussed this limitation more explicitly, adding further explanation to the Methods and Discussion.

Note that chamber-related problems (e.g. aphid infestations) primarily occurred under warm chamber conditions, whereas our experiment 1 cooling treatments maintained low temperatures that suppressed such issues. This means that an equivalent “warm chamber control” could have been associated with its own artefacts, as trees kept under warm chamber conditions would have been exposed to additional stressors that were not present under natural growing conditions. To address this point, we included a chamber control in experiment 2. While aphid abundance was indeed higher in the warm chamber controls, chamber exposure itself had no detectable effect on autumn phenology. This suggests that the main findings of experiment 1 are unlikely to be artefacts of chamber conditions.

Nevertheless, we agree that chamber exposure remains a potential limitation of experiment 1, which requires clear acknowledgement. We now state this more explicitly in the manuscript while also emphasising that our results are supported by experiment 2 and by converging lines of external evidence.

Also, I suggest the authors add a figure to explain their experiments as they are very hard to follow. Perhaps this could be added to Figure 1?

We have now added figures to the methods section to depict the experimental timelines and settings more clearly (Figs. 2 and 3).

Finally, given how much the authors extrapolate to carbon and forests, I would have liked to see some metrics related to carbon assimilation, versus just information on timing.

We agree that carbon assimilation is an important component of forest carbon dynamics. However, the primary aim of this study was to identify how developmental state and diel cycles mediate temperature effects on autumn phenology, rather than to quantify carbon assimilation per se. Assessing photosynthetic controls on autumn phenology would require a substantially different experimental design and is therefore beyond the scope of the present study.

That said, we were able to include measurements of photosynthetic assimilation during pre-solstice cooling (now presented as Fig. S12 for all treatments). These data show that cooling strongly reduced assimilation across all treatments, despite their markedly different phenological outcomes. This supports our interpretation that variation in assimilation alone cannot explain the observed phenological responses, consistent with previous manipulative and observational studies reporting a weak role of late-season assimilation in controlling autumn phenology.

Fagus sylvatica: Fagus sylvatica is an extremely important tree to European forests, but it also has outlier responses to photoperiod and other cues (and leafs out very late) so using just this species to then state 'our results likely are generalisable across temperate tree species' seems questionable at best.

We agree that Fagus sylvatica has a stronger photoperiod dependence than many other European tree species. As we note in our response to Reviewer 1, our findings align with previous research across temperate northern forests. Within our framework, interspecific variation in leaf-out timing would not alter the overall response pattern, though it could shift the specific timing of effect reversals. For example, earlier-leafing species may approach completion of development sooner and thus show sensitivity to late-season cooling earlier than F. sylvatica. Nevertheless, we acknowledge the importance of not overstating generality. We have therefore revised the manuscript to phrase conclusions more cautiously and highlight the need for further research across species.

And the referenced response to Reviewer one:

We agree that extrapolation from our experiments on Fagus sylvatica to other species and natural forests requires caution. However, it is precisely the controlled nature of our design that allowed us to isolate the precise mechanisms that appear to underpin the solstice switch, highlighting the role of diel and seasonal temperature variation. In natural systems, additional variables such as competition, precipitation, and soil heterogeneity can strongly influence phenology, but they also make it difficult to disentangle causal mechanisms. By minimising these confounding factors, our experiment provided a clear test of how temperature before and after the solstice regulates growth cessation.

To acknowledge the limitation, we have toned down statements about generalisation (e.g. “likely generalisable” to “other temperate tree species may display similarities”) and explicitly call for follow-up studies across species and forest contexts. At the same time, we highlight that our findings align with independent evidence from manipulative experiments, satellite observations, flux measurements, and groundbased phenology, which suggests the mechanisms we report may extend beyond the specific populations studied here.”

As described in responses above, we have further clarified what can be directly concluded from our study, avoiding overgeneralisation.

Measuring end of season (EOS): It's well known that different parts of plants shut down at different times and each metric of end of season -- budset, end of radial expansion, leaf coloring etc. -- relate to different things. Thus I was surprised that the authors ignore all this complexity and seem to equate leaf coloring with budset (which can happen MONTHS before leaf coloring often) and with other metrics. The paper needs a much better connection to the physiology of end of season and a better explanation for the focus on budset. Relatedly, I was surprised the authors cite almost none of the literature on budset, which generally suggests is it is heavily controlled by photoperiod and population-level differences in photoperiod cues, meaning results may different with a different population of plants.

We thank the reviewer for pointing out that our discussion of the responses of different EOS metrics needs more clarity. We agree with much of this perspective, and we have added an additional analysis of leaf chlorophyll content data to use leaf discolouration as an alternative EOS marker. On this we would like to make two important points:

Firstly, we agree that bud set often occurs before leaf discolouration, although this can depend on which definition of leaf discolouration is used. In experiment 1, budset occurred on average on day-of-year (DOY) 262 and leaf senescence (50% loss of leaf chlorophyll) occurred on DOY 320. However, we do not necessarily agree that this excludes the combined discussion of bud set and leaf senescence timing. Whilst environmental drivers can affect parts of plants differently, often responses from different end-of-season indicators (e.g. bud set and loss of leaf chlorophyll) are similar, even if only directionally. Figure S11 shows how, across both experiments, treatment effects were tightly conserved (R^2^ = 0.49) amongst the two phenometrics. In accordance with these revisions, we have updated the manuscript title to “Developmental constraints mediate the summer solstice reversal of climate effects on the autumn phenology of European beech”.

Secondly, shifts in bud set timing remain the primary focus of the manuscript as these shifts are of direct physiological relevance to plant development and dormancy induction, whereas leaf discolouration may simply follow bud set as a symptom of developmental completion. This is supported by our results, which show stronger responses of bud set than leaf senescence (Figs. 4 & 5 vs. Figs. S9 & S10).

Following the reviewer’s suggestion, we have included more references on the topic of bud set and its environmental controls. The reviewer rightly stresses that photoperiod is considered the most important factor. Photoperiod is therefore key in our conceptual model. However, the responses we observed in F. sylvatica cannot be explained by photoperiod alone. For example, in experiment 1, July cooling delayed the autumn phenology of late-leafing trees but had negligible impact on early-leafing trees, even though both experienced the exact same photoperiod. Moreover, in experiment 2, day, night and full-day cooling showed substantial variations in their effects despite equal photoperiod across the climate regimes. This is why we suggest that the annual progression of photoperiod modulates the responses to temperature variations instead of eliciting complete control.

Following the addition of an analysis of leaf senescence data, we also revised the terminology in places (including the title) from “primary growth cessation/bud set” to the broader term “autumn phenology.” This term is intended to encompass two distinct but related physiological processes—bud set and leaf senescence—both of which are commonly used as markers of autumn phenology and the end of the growing season.

Somewhat minor comments:(1) How can a bud type -- which is apical or lateral -- be a random effect? The model needs to try to estimate a variance for each random effect so doing this for n=2 is quite odd to me. I think the authors should also report the results with bud type as fixed, or report the bud types separately.

We have revised the analysis to include bud type as a fixed effect. There are only very minor numerical adjustments (e.g. rounding to 4.8 days instead of 4.9) and inferences are not altered. We also report the bud type effects for experiment 1 and experiment 2.

(2) I didn't fully see how the authors results support the Solstice as Switch hypothesis, since what timing mattered seemed to depend on the timing of treatment and was not clearly related to solstice. Could it be that these results suggest the Solstice as Switch hypothesis is actually not well supported (e.g., line 135) and instead suggest that the pattern of climate in the summer months affects end of season timing?

Our responses to the main comments in this new round of revision have comprehensively covered this topic.

References

Berend K, Haynes K, MacKenzie CM. 2019. Common garden experiments as a dynamic tool for ecological studies of alpine plants and communities in northeastern North America. Rhodora 121: 174.

Heide OM. 2008. Interaction of photoperiod and temperature in the control of growth and dormancy of Prunus species. Scientia Horticulturae 115: 309–314.

Körner C. 2021. Alpine Plant Life: Functional Plant Ecology of High Mountain Ecosystems. Cham: Springer International Publishing.

Somero GN. 2010. The physiology of climate change: how potentials for acclimatization and genetic adaptation will determine ‘winners’ and ‘losers’. Journal of Experimental Biology 213: 912–920.

Tanino KK, Kalcsits L, Silim S, Kendall E, Gray GR. 2010. Temperature-driven plasticity in growth cessation and dormancy development in deciduous woody plants: a working hypothesis suggesting how molecular and cellular function is affected by temperature during dormancy induction. Plant Molecular Biology 73: 49–65.